# Polycistronic Genome Segment Evolution and Gain and Loss of FAST Protein Function during Fusogenic Orthoreovirus Speciation

**DOI:** 10.3390/v12070702

**Published:** 2020-06-29

**Authors:** Yiming Yang, Gerard Gaspard, Nichole McMullen, Roy Duncan

**Affiliations:** 1Department of Microbiology and Immunology, Dalhousie University, Halifax, NS B3H 4R2, Canada; yiming.yang@dal.ca (Y.Y.); Gerard.gaspard@dal.ca (G.G.); nichole.mcmullen@dal.ca (N.M.); 2Department of Biochemistry and Molecular Biology, Dalhousie University, Halifax, NS B3H 4R2, Canada

**Keywords:** orthoreovirus, FAST proteins, syncytium formation, recombination, virus evolution

## Abstract

The *Reoviridae* family is the only non-enveloped virus family with members that use syncytium formation to promote cell–cell virus transmission. Syncytiogenesis is mediated by a fusion-associated small transmembrane (FAST) protein, a novel family of viral membrane fusion proteins. Previous evidence suggested the fusogenic reoviruses arose from an ancestral non-fusogenic virus, with the preponderance of fusogenic species suggesting positive evolutionary pressure to acquire and maintain the fusion phenotype. New phylogenetic analyses that included the atypical waterfowl subgroup of avian reoviruses and recently identified new orthoreovirus species indicate a more complex relationship between reovirus speciation and fusogenic capacity, with numerous predicted internal indels and 5’-terminal extensions driving the evolution of the orthoreovirus’ polycistronic genome segments and their encoded FAST and fiber proteins. These inferred recombination events generated bi- and tricistronic genome segments with diverse gene constellations, they occurred pre- and post-orthoreovirus speciation, and they directly contributed to the evolution of the four extant orthoreovirus FAST proteins by driving both the gain and loss of fusion capability. We further show that two distinct post-speciation genetic events led to the loss of fusion in the waterfowl isolates of avian reovirus, a recombination event that replaced the p10 FAST protein with a heterologous, non-fusogenic protein and point substitutions in a conserved motif that destroyed the p10 assembly into multimeric fusion platforms.

## 1. Introduction

The fusogenic reoviruses are rare examples of non-enveloped viruses that induce cell–cell fusion and syncytium formation [1]. This diverse group of non-enveloped viruses, with segmented, dsRNA genomes, are members of two distinct but related genera in the *Reoviridae* family: the *Orthoreovirus* genus, whose members infect a wide range of vertebrate hosts, and the *Aquareovirus* genus, whose hosts are restricted to freshwater and saltwater fish [2]. Three of the five recognized and sequenced species in the *Aquareovirus* genus are fusogenic. Of the seven currently recognized species of orthoreoviruses, only *Mammalian orthoreovirus* (MRV) and *Piscine orthoreovirus* (PRV) lack isolates that are fusogenic. Isolates of *Reptilian orthoreovirus* (RRV), *Mahlapitsi orthoreovirus*, (MaRV), *Nelson Bay orthoreovirus* (NBV), *Baboon orthoreovirus* (BRV), and *Avian orthoreovirus* (ARV) all induce syncytium formation [3,4]. Furthermore, three additional orthoreovirus species have been proposed, all of which are fusogenic: *Broome orthoreovirus* (BrRV) from a bat, *Neoavian orthoreovirus* (ARVN) from wild birds, and *Testudine orthoreovirus* (RRVT) from a tortoise (Figure 1A). The preponderance of fusogenic over non-fusogenic reoviruses suggests there is positive evolutionary pressure to obtain and/or maintain the fusogenic phenotype.

Orthoreoviruses are members of the *Spinareovirinae* subfamily, whose virions have large turrets at the icosahedral vertices. The double-layered capsid contains eight structural proteins, five of which comprise the transcriptionally active core particle containing the viral RNA-dependent RNA polymerase (RdRp) [5,6]. Two of the three remaining proteins, the outer clamp and outer shell proteins, assemble into heterohexameric complexes and form the outer capsid shell [7]. During reovirus endocytic cell entry, cathepsins proteolytically degrade the outer clamp protein to expose the underlying outer shell protein that mediates the exit of the core particle from the endocytic pathway into the cytoplasm [8]. The RdRp functions from within the core particle to transcribe mostly monocistronic mRNAs from the 10 genome segments that are capped but not polyadenylated by the turret protein during the exit from the core particle [9,10]. At some stage, interactions between the viral RdRp, packaging signals in the viral mRNA, and the core shell protein result in the coordinated replication of the mRNA and the encapsidation of the 10 dsRNA genome segments inside the progeny virus core particles [11,12]. These transcriptionally active progeny core particles drive mRNA synthesis and amplification of the virus replication cycle. The three outer capsid proteins eventually assemble on the core particles to generate infectious progeny virions. During the progeny core assembly stage of the replication cycle in cells which have been co-infected with two different reoviruses, heterologous viral mRNAs can be encapsidated and replicated inside a single progeny core particle to generate reassortant viruses. The genomes of these reassortants are mosaics of the parental virus genome segments and, as with antigenic shift in influenza viruses [13], genome segment reassortment is a major driver of reovirus evolution [14].

The final structural protein is the fiber protein, referred to as σ1 in MRV and σC in the fusogenic reoviruses, that forms trimeric spikes at the capsid vertices and serves as the cell adhesin [15,16,17]. The reovirus fiber protein comprises a long N-terminal filamentous tail and an ~140-residue C-terminal globular head that folds into a compact, eight-stranded B-barrel structure with receptor binding activity [18,19]. The filamentous tail contains three structurally distinct subregions: the N-terminal 20–30 residues interact with the turret protein to anchor the spike to the virion capsid; a long α-helical coiled coil domain formed by heptad repeats zips together the trimeric fiber; and a region of triple β-spiral repeats connects the tail to the head [20,21,22]. In those fusogenic reoviruses that encode a fiber (Figure 1B), the σC fiber is shorter than the MRV σ1 fibers (~320 versus 455–470 residues, respectively), due primarily to a reduced number of β-spiral repeats [19]. In some reovirus species (MaRV, BRV, BrRV), the fiber protein ORF has been completely replaced by a heterologous sequence encoding a non-structural p16 protein of no defined function (Figure 1B). It is unclear how these fiberless virus particles attach to cell receptors.

The genomes of all fusogenic aqua- and orthoreoviruses contain an additional open reading frame (ORF) which is not found in their non-fusogenic counterparts. This ORF is present at the 5’end of a bi- or tricistronic mRNA and encodes the fusion-associated small transmembrane (FAST) protein responsible for cell–cell fusion and syncytium formation (Figure 1B) [1]. FAST proteins are a unique family of viral membrane fusion proteins that are structurally and evolutionarily unrelated to the fusion proteins that are responsible for enveloped virus entry into cells. These “accessory” proteins are non-essential for productive virus infection and they are non-structural proteins, hence they are not involved in cell entry. FAST proteins are small, integral membrane proteins expressed inside reovirus-infected cells where they localize at adherens junctions and mediate cell–cell, not virus–cell, membrane fusion [23]. In so doing, FAST proteins promote the rapid cell–cell transmission of the infection, a possible explanation for the positive selection pressure on acquiring or maintaining a FAST protein. Natural fusogenic reovirus infections are frequently associated with overt disease. ARV is responsible for outbreaks of enteritis, tenosynovitis, and myocarditis in commercial poultry flocks [24], RRV is associated with pneumonia and neurological dysfunctions in snakes [25,26], BRV causes meningoencephalomyelitis in nonhuman primates [27,28], and NBV has been associated with acute respiratory infections in humans. Studies in different experimental systems link FAST proteins to the pathogenic potential of these viruses [29,30,31,32].

There are currently six members of the aqua- and orthoreovirus FAST protein family that share limited sequence conservation but share several hallmark biochemical features [1,33]. The remainder of this discussion will focus on the orthoreovirus FAST proteins (Figure 1C). All FAST proteins are small (~100–200 residues), acylated (shown or predicted to be palmitoylated or myristoylated), single-pass membrane proteins that position a very small (<40 residues) N-terminal ectodomain external to the plasma membrane and an extended (~40–140 residues) C-terminal endodomain in the cytoplasm [34,35,36]. FAST proteins appear to have evolved via the assembly of diverse membrane interaction motifs into three small domains (ecto-, endo-, and transmembrane) that function as fusion modules. The ectodomains are amphiphilic, structurally dynamic peptides, with remarkably diverse structures, that function as fusion peptides to destabilize lipid bilayers [3,29,37,38,39]. FAST protein TM domains contain different motifs, adjacent to the ectodomain-transmembrane domain boundary, that are required for cell–cell fusion (e.g., polyglycine in p10, polyserine and hydrophobic β-branched residues in p15) [36,40,41]. The FAST protein endodomains all contain a juxtamembrane cluster of basic residues and an adjacent amphipathic helix that are essential for syncytium formation. The polybasic motif functions as a Golgi export signal for the RRV p14 FAST protein [42,43], while the amphipathic helices in the p14 and BRV p15 FAST proteins partition into highly curved membranes to promote pore formation [44]. The C-terminal half of the FAST protein endodomains are predicted to be disordered [45], a common feature of protein regions that interact with multiple partners; annexin A1 is one such p14 partner [46]. Lastly, FAST proteins are modular fusogens, meaning that the exchange of modules between different FAST proteins can generate functional recombinants, although not all combinations are tolerated. The noteworthy mechanistic implication of the above discussion is that specific combinations of different structural motifs can be assembled in order to generate a functional FAST protein [3,40].

The modular nature of FAST proteins and the diversity of the functional motifs that they employ to drive membrane fusion raises interesting questions regarding the evolution of these non-enveloped virus fusogens. Previous phylogenetic studies that included the six available sequenced species of orthoreoviruses suggested two separate gain-of-fusion events caused by a common ancestral, non-fusogenic reovirus led to the extant fusogenic aqua- and orthoreoviruses [2]. The evolutionary relationship between the orthoreovirus FAST proteins was unclear. These studies also overlooked the waterfowl subgroup of ARV typified by the Muscovy duck reoviruses (MdRV). The MdRV infection of young ducklings results in high morbidity and mortality (10%–50% mortality), with necrotic focus formation in the liver, spleen, and kidneys [47]. Early isolates of MDRV were reported to be syncytiogenic like their landfowl counterparts [47], but no sequence information is available for these isolates. All the more recent MdRV isolates lack syncytium-inducing capabilities [48,49]. MdRVs are divided into two subgroups, the “classical” and “novel” MdRVs. Classical Muscovy duck reoviruses (MdRVc) possess a bicistronic S4 genome segment encoding a truncated fiber protein of 269 residues and a p11 protein that lacks sequence similarity to the p10 FAST proteins of ARV, ARVN, and NBV (Figure 1B) [50]. The more recently identified novel subgroup (MdRVn) has a tricistronic S1 genome segment similar to other ARVs [48], encoding a p10 FAST protein homologue, a p18 protein of unknown function, and a 326-residue fiber protein (Figure 1B). There has been little discussion in the literature regarding the loss of fusion activity in MdRVs or the genetic events underlying the evolution of the orthoreovirus polycistronic genome segments.

As we now show, using phylogenetic analysis that includes the current seven orthoreovirus species, three new proposed species, and previously excluded non-fusogenic MdRVs, the evolution of the orthoreovirus polycistronic genome segment from an inferred ancestral, monocistronic fiber-encoding genome segment involved polymorphic, species-specific indels at a “genetic hotspot” in the fiber ORF. These genome segments further evolved by the addition of 5’-terminal extensions that added additional ORFs upstream of the fiber ORF, one of which encoded a FAST protein homologue. Evidence suggests these 5’-terminal recombination events occurred multiple times, both pre- and post-speciation, to generate the constellation of extant orthoreovirus polycistronic genome segments and FAST proteins. We further directly demonstrate that point substitutions in a nine-residue multimerization motif are solely responsible for the loss of fusion capability in MdRVn isolates. These results highlight the underappreciated role of recombination, not just genome segment reassortment, in reovirus evolution, and support the concept that FAST proteins evolved from small ancestral membrane proteins by independently acquiring the correct assembly of diverse membrane interaction motifs needed to generate a fusogenic FAST protein.

## 2. Materials and Methods

### 2.1. Cells and Reagents

QM5 cells were maintained in medium 199 with Earle’s salts containing 100 U/mL of penicillin and streptomycin and 10% heat-inactivated fetal bovine serum. Monoclonal mouse anti-FLAG (Sigma-Aldrich) and rabbit anti-Myc (Sigma-Aldrich) antibodies, horseradish peroxidase (HRP)-conjugated goat anti-rabbit (Santa Cruz) and goat-anti mouse (Santa Cruz) antibodies, Alexa Fluor 488-conjugated goat anti-mouse (Invitrogen) and Alexa Fluor 555-conjugated goat anti-rabbit (Invitrogen), Sulfo-NHS-LC-Biotin (Thermo Scientific), maliemide-PEG2-biotin (Thermo Scientific), neutravidin agarose resin (Thermo Scientific), and polyethylimine (PEI, Polysciences Inc.) were purchased from the indicated commercial sources.

### 2.2. Plasmids and Cloning

ARV p10, MdRV p10, and Md-CM (MdRV p10 containing the 9-residue conserved motif of ARV p10 created using sequential PCR reactions with custom oligonucleotide primers) were subcloned into pcDNA3 mammalian expression vectors between the HindIII and EcoRI sites. A triple FLAG or single Myc tag was added to the N-terminus, or EGFP or mCherry were added to the C-terminus of the indicated p10 constructs by PCR amplification and cloning. Custom oligonucleotide primers were purchased from Integrated DNA Technologies (IDT), and all constructs were confirmed by sequencing.

### 2.3. Transfections and Syncytial Indexing

QM5 cell monolayers at 50% confluency in 12-well plates were transfected with 0.5 µg of plasmid DNA using PEI as per the manufacturer’s instructions. At 4 h post-transfection, the transfection mix was replaced with medium 199 (Gibco), supplemented with 10% fetal bovine serum (Sigma-Aldrich). At the indicated times post-transfection, cells were fixed with methanol and Wright–Giemsa stained (Siemens Healthcare Diagnostics), and the stained monolayers were imaged using a Nikon DIAPHOT microscope at 100× magnification. The numbers of syncytial nuclei were counted in five random fields of triplicate wells (*n* = 3 independent experiments) and the syncytial index reported as the mean ± SEM.

### 2.4. SDS-PAGE and Western Blotting

SDS-PAGE and Western blotting were carried out as previously described [17,49]. A 1:1000 dilution of primary anti-FLAG or anti-Myc antibodies, followed by a 1:10,000 dilution of HRP-conjugated secondary antibody was used to measure overall protein expression levels. Western blots were developed using Clarity Western ECL substrate (Bio-Rad) and imaged on a ChemiDoc Imaging System (Bio-Rad).

### 2.5. Cell Surface Immunofluorescence Microscopy

QM5 cells grown to approximately 50% confluence on coverslips were transfected with FLAG- or Myc-tagged versions of ARV or MdRV p10 constructs, as described above. At 24 h post-transfection, cells were fixed with 3.7% paraformaldehyde for 10 min at room temperature, blocked with 5% bovine serum albumin (BSA) in phosphate buffered saline (PBS) for 1 h, and then incubated with 1:200 dilutions of mouse anti-FLAG and rabbit anti-Myc monoclonal antibodies overnight at 4 °C. Cells were thoroughly washed with PBS and then incubated with 1:1000 dilutions of Alexa 488-conjugated goat anti-mouse antibody and Alexa 555-conjugated goat anti-rabbit antibody for 1 h at room temperature. Coverslips were then mounted using prolong gold anti-fade reagent (Invitrogen) and imaged using a Zeiss LSM510 Axiovert 200M confocal microscope.

### 2.6. Cell Surface Biotinylation

QM5 cells grown in 10 cm dishes (Corning) to 50% confluency were transfected with the indicated FLAG-tagged constructs. At 24 h post-transfection, cells were washed three times with ice-cold PBS (pH 8.0), then incubated with shaking in 1 mg/mL Sulfo-NHS-LC-Biotin (a membrane impermeable biotinylation reagent) for 2 h at 4 °C to biotinylate primary amino groups on the cell surface. Cells were washed with PBS containing 100 mM glycine to quench and remove excess biotin reagent. Cells were washed twice with PBS, then lysed in a RIPA buffer (10 mM Tris pH 8.0, 150 mM NaCl, 1 mM EDTA, 1% NP40, 0.5% Na desoxycholate) with protease inhibitors (Pierce). Cell lysates were incubated overnight with neutravidin agarose resin to pull down the biotinylated proteins, and pellets were boiled in a protein sample buffer with 100 mM dithiothreitol (DTT) to release the biotinylated proteins. Samples were then analyzed by SDS-PAGE and Western blotting.

To detect the presence of an intramolecular disulfide bond in the p10 ectodomain, QM5 cells transfected with Myc-tagged p10 constructs were washed twice with Hanks buffered saline solution (HBSS) at 24 h post-transfection and then incubated in HBSS with or without 0.1 mM DTT for five min. Cells were washed three times with HBSS, then incubated with shaking in 1 mg/mL maliemide-PEG2-biotin (a membrane impermeable biotinylation reagent) for 25 min at 4 °C to biotinylate free thiol groups on the cell surface. Cells were washed four times with HBSS to remove excess biotin reagent and once with HBSS containing 1% BSA to quench residual biotinylation reagent. Cells were washed twice with PBS, resuspended with 50 mM EDTA in PBS, and lysed in the RIPA buffer with protease inhibitors, and the lysate was incubated overnight with neutravidin agarose resin to pull down biotinylated proteins. Pellets were boiled in a protein sample buffer with 100 mM DTT to release biotinylated proteins and analyzed by SDS-PAGE and Western blotting.

### 2.7. FRET-Based Multimerization Assay

A Zeiss LSM510 confocal microscope was used in wide field mode to detect sensitized emission fluorescence resonance energy transfer (FRET) from QM5 cells transfected with EGFP- and/or mCherry-tagged versions of the indicated p10 constructs, as previously described [26,51]. Briefly, donor and acceptor spectral bleed-through (SBT) values were minimized using cells expressing EGFP or mCherry alone, and donor and acceptor SBT ratios were modeled using exponential relationships with fluorophore intensity, after the exclusion of aberrant background values at low intensities and the application of a Gaussian blur. Using this microscope setup and the SBT values, three images were acquired from each cell that was imaged: (1) a donor image (donor excitation, donor emission), (2) an acceptor image (acceptor excitation, acceptor emission), and (3) a sensitized emission FRET image (donor excitation, acceptor emission). The background subtraction and Gaussian blur of the donor, acceptor, and FRET channels were performed on each image stack prior to analysis. Ten images were acquired for each sample from each of the two independent experiments (total of twenty images per sample). The PixFRET Image J plugin [46,52,53] was used to normalize the FRET intensity of each pixel to the donor and acceptor expression levels by dividing the FRET-channel pixel intensity by the square-root of the product of the corresponding donor- and acceptor-channel pixels, using the following equation:$$NFRET=[\frac{I_{FRET}-I_{EGFP}\times BT_{EGFP}-I_{mCHerry}\times BT_{mCherry}}{\sqrt{I_{EGFP}\times I_{mCHerry}}}]$$

The Gaussian distribution with the highest calculated R^2^ value for the pixel amplitude distributions generated by PixFRET from the 8-bit normalized FRET (NFRET) images was used to calculate the mean NFRET (mNFRET) for each image.

### 2.8. Phylogenetic Analysis

Phylogenetic analysis used Clustal Muscle to generate multiple sequence alignments of homologous proteins from the different orthoreovirus species that were analyzed for evolutionary relationships, using MEGA 7.0 and the maximum likelihood method based on the Jones-Taylor-Thornton (JTT) matrix-based model [54,55]. These methods were applied to concatenated protein sequences for the seven homologous orthoreovirus structural proteins from either the 10 orthoreovirus species or the proposed species (excluding the highly variable fiber protein), to the sigma-class outer capsid clamp proteins of either the eight fusogenic orthoreovirus species or the proposed species, and to the FAST proteins encoded by these eight virus species. Accession numbers are provided in the figures or figure legends and in the Appendix A.

## 3. Results and Discussion

### 3.1. Evolution of the Fiber-Encoding Genome Segment Involved Internal Indels in the Fiber ORF

Phylogenetic analysis revealed species-specific features of the different orthoreovirus fiber proteins and the polycistronic genome segments encoding these proteins. The orthoreovirus fiber proteins share the same overall trimeric structure but with considerable variation in their lengths, ranging from 269 to 470 residues (Figure 2A). All orthoreovirus fiber-encoding genome segments are polycistronic except the PRV S4 genome segment. The PRV genome segment is ~400 nucleotides shorter than the corresponding S1 genome segment of MRV due to an internal indel of ~400–550 nucleotides within the region of the fiber ORF where the coiled-coil tail joins the β-spiral repeats (Figure 2A). Consequently, PRV contains 17–18 predicted heptad repeats in its coiled coil tail and only two β-spiral repeats while MRV fibers contain 20–21.5 heptads and seven β-spiral repeats [18,19,21,56]. Viral polymerase errors frequently generate substitutions, but indels also occur at about a four-fold lower frequency [57]. Excluding MRV, all other orthoreovirus species, including those whose hosts are more ancient (i.e., fish, reptiles, and birds), contain homologous short versions of the trimeric fiber. The logical inference is the ancestral fiber genome segment resembled that of PRV, encoding a short fiber on a monocistronic mRNA.

The fiber ORF indels are polymorphic but species-specific in their length, implying indels altered the fiber protein several times during orthoreovirus speciation. The extant fiber proteins cluster into four groups based on their total lengths and the approximate number of heptad and β-spiral repeats: the PRV/ARV/MdRVn/ARVN/NBV fiber group (17–18 predicted heptad and two predicted β-spiral repeats), the MdRVc fiber protein (10 predicted heptad and two predicted β-spiral repeats), the RRV/RRVT clade (19 predicted heptad and three potential β-spiral repeats), and the MRV fibers (20–21.5 heptads and seven β-spiral repeats) (Figure 2A) [49,50]. Three error-prone polymerase events, giving rise to indels with variable lengths, primarily during the synthesis of the β-spiral repeat region, would explain the evolution of the PRV/ARV/MdRVn/ARVN/NBV fiber protein into the extant MRV, RRV/RRVT, and MdRVc fiber proteins (Figure 2B). It would appear that this structural motif is highly tolerant to changes in the number of repeats without losing protein function, namely the cell attachment capacity. As occurs with recombination events in many RNA viruses [58], undefined RNA secondary structures in this region might also play a role in making this region a hotspot for such genetic events. Sequence analysis suggests these fiber ORF indels continue to occur, as evidenced by an additional 18-residue indel detected in the fiber ORF of a recent MdRVn isolate that creates a 303-residue MdRVn fiber protein lacking an additional β-spiral repeat [59].

### 3.2. Acquisition of FAST Protein Precursors by 5’-Terminal Extensions of an Ancestral Fiber-Encoding Genome Segment 

The fiber ORF indels are one major contributor to the variable lengths of the different polycistronic genome segments. The second is what can only be inferred are 5’-terminal extensions, assuming that the last common ancestor of the extant fiber-encoding genome segments resembled that of PRV. These ~400–700 nucleotide extensions in the fusogenic orthoreoviruses added additional reading frames upstream of the fiber ORF, one of which encoded a FAST protein homologue (Figure 2A). Template-switching as the RdRp replicates the plus-strand into a minus-strand would generate such genetic hybrids with 5’-extensions, as previously reported in plant viruses [60]. This copy choice mechanism of recombination is common in many plant viruses and in plus-strand RNA animal viruses including retroviruses, enteroviruses, and coronaviruses, and may simply reflect the stochastic association and dissociation of the RdRp from the RNA template [58,61]. RNA recombination usually occurs between regions of high sequence similarity. Such homologous recombination between cognate genome segments has been noted in ARVs, where as many as twelve recombination events between multiple cognate genome segments of different circulating virus strains have been predicted [62]. However, non-homologous recombination can also occur, with genetic exchange taking place between different virus genomes [63] or, more rarely, between viral and cellular RNA [64,65,66]. The high degree of sequence divergence in the polycistronic orthoreovirus mRNAs and the likelihood that ancient, non-homologous recombination events generated these 5’-terminal extensions confounds similarity searches to define specific breakpoints or even the potential source of the alternate template RNA. However, additional analyses discussed below support the concept that such genetic events occurred more than once and were major evolutionary drivers of the fusogenic orthoreovirus polycistronic genome segments and the FAST proteins that they encode.

### 3.3. Extant p10 FAST Proteins Arose by an Independent Recombination Event

The distinct features of the p10 FAST proteins and their tricistronic genome segments compared to the other orthoreovirus FAST proteins and their bicistronic genome segments imply that at least two distinct recombination events occurred during the fusogenic reovirus evolution. Despite their distinct host ranges (i.e., birds and bats), ARV, ARVN, and NBV form a monophyletic clade (Figure 3A) and they share a tricistronic gene arrangement comprising three sequential, partially overlapping ORFs encoding a p10 FAST protein, a non-structural p17 protein, and the σC fiber protein (Figure 3B). This includes the MdRVn, but not the MdRVc, subgroup of waterfowl ARV isolates; the latter have a bicistronic gene arrangement and are further discussed below. The ARV, ARVN, and NBV p10 FAST proteins also form a monophyletic clade that is distinct from the p13, p14, and p15 FAST proteins, with a 94% bootstrap value at that branchpoint (Figure 3C). In multiple sequence alignments, the p10 FAST proteins of the three species (ARV, ARVN, and NBV) are essentially colinear (two small, terminal indels) and share ~38%–65% amino acid identity in pairwise comparisons (Figure 4A). This is not the case for alignments of p10 with the other FAST proteins, where ~20 indels are needed to attain amino acid identities of only ~13%–23% (data not shown). The p10 proteins also contain several conserved features which are not found in the other orthoreovirus FAST proteins—an unusual ectodomain cystine noose fusion peptide, a 9-residue ectodomain conserved motif, a palmitoylated endodomain di-cysteine motif instead of N-terminal myristoylation, and a short cytoplasmic tail (Figure 1C) [36,51,52,67]. Based on the above considerations, we infer that the predicted 5’-extension that led to the evolution of the p10 FAST proteins occurred after the RRV/BrRV/BRV/MaRV and ARV/NBV orthoreovirus clades diverged from each other (Figure 3A).

In addition to p10, the ARV/ARVN/NBV clade acquired a second ORF upstream of the fiber ORF encoding a p17/p18 protein. The ARV p17 protein is a nucleocytoplasmic shuttling protein that suppresses the cell cycle ([68,69], but similar attributes have not been ascribed to other reovirus p17 proteins or to the MdRVn p18 protein. Alignments of the p17 proteins from the NBV, ARVN, and landfowl isolates of ARV are mostly colinear (Figure 4B), and in pairwise comparisons of the more distantly related species these proteins still share ~23%–33% amino acid identity which is suggestive of a homologous relationship. However, this degree of amino acid conservation is considerably lower than that between the NBV and ARV/ARVN p10 proteins (~38%–65% identity). The p18 protein of MdRVn shares even less similarity with p17; protein alignments require 7–8 internal indels, totaling >100 nucleotides, in order to generate only ~12%–20% amino acid identity (data not shown). It is therefore unclear whether one recombination event simultaneously added precursors of the p10/p17/p18 ORFs that then evolved at very different rates or independent recombination events added these ORFs separately (Figure 3A).

### 3.4. Evolution of the Fusogenic Bicistronic Genome Segments Involved in Independent Recombination Events and Lateral Gene Transfer

The BrRV, MaRV, RRV, RRVT, and BRV virus species form a monophyletic clade, as do their encoded p13, p14, and p15 FAST proteins (Figure 3A–C). These three diverse FAST proteins are encoded by the first ORFs of bicistronic genome segments that have two distinct gene arrangements. All reptilian orthoreovirus isolates, including the tortoise isolate of the new proposed RRVT species, encode homologous p14 FAST proteins using the first ORF and retain a truncated σC fiber protein ORF (Figure 3B). BRV, BrRV, and MaRV encode p13, p14, or p15 FAST proteins, but their fiber ORFs have been replaced by an unrelated ORF encoding a p16 protein of no defined function. Aside from terminal indels to accommodate length differences, these p16 proteins are mostly colinear (four single residue indels) and share 28%–30% amino acid identity clustered in subregions (Figure 4C), indicating a shared common ancestor. BRV, MaRV, and BrRV form a monophyletic clade when comparing their sigma-class outer clamp capsid proteins (Figure 3A) or lambda-class core shell proteins [70], consistent with a single recombination event replacing the fiber ORF with the p16 ORF prior to speciation of these viruses. Since the three viruses are paraphyletic when comparing other homologous proteins (Figure 1A), lateral gene transfer via genome segment reassortment between these evolving virus species presumably occurred after the p16 recombination event. Recombination events and genome segment reassortment are well-linked to intra-species ARV evolution, with some suggestion that similar inter-species genetic events may have occurred between orthoreovirus species [62], possibly early post-speciation.

A single recombination event that extended the 5’- end of the fiber ORF with an ORF encoding a shared ancestral protein could explain the evolution of the BrRV, MaRV, RRV, RRVT, and BRV p13, p14, and p15 FAST proteins (Figure 3A). Evidence does support the divergent evolution of the BrRV p13 and reptilian reovirus p14 FAST proteins from a shared, fusogenic ancestor. Phylogenetic trees based on FAST protein sequences have weak bootstrap values at branchpoints separating the single isolates of BRV p15, MaRV p14, and BrRV p13, making evolutionary relationships unclear (Figure 3C). However, the p14 FAST proteins of RRV, RRVT, and MaRV are clear homologues. These proteins share conserved sequence motifs, including a conserved N-terminal decapeptide that is sensitive to substitution and contains the N-terminal myristoylation consensus sequence (Figure 4D) [35], and they share ~27%–36% amino acid identity. BrRV p13 FAST protein possesses the same motifs as p14, including the N-terminal decapeptide but excluding a cytoplasmic PPXY motif of no defined function (Figure 4D), and in mostly colinear, pairwise sequence alignments it shares ~22%–29% sequence identity with MaRV and RRV p14. The hypothesis that the p13 and p14 FAST proteins are homologues is consistent with all phylograms that show a monophyletic relationship between BrRV, MaRV, RRV, and RRVT and their encoded p13 and p14 FAST proteins (Figure 1A and Figure 3A).

As indicated by the FAST protein and concatenated sequence phylograms, BRV and the p15 FAST protein present as outliers. Pairwise alignments of p15 with p14 or p13 require 7–8 large indels, totaling ~50–80 residues, to generate ~20%–24% sequence identity. A similar level of sequence conservation (~23% amino acid identity in alignments containing four large indels) is observed between BRV p15 and a recently identified p11 FAST protein (referred to as NSP1-1, the first ORF on a bicistronic mRNA) of rotavirus species B (RVB), a virus from a distantly related genus and different subfamily of *Reoviridae* [71]. The BRV p15 FAST protein also contains several features not associated with p13 or p14; p15 lacks the conserved N-terminal decapeptide sequence present in p13 and p14, but it does contain a functional GXXXS myristoylation site [34,72]. The ~20-residue ectodomain is markedly truncated and contains a unique polyproline motif that forms a left-handed polyproline type II helix required for fusion activity [38], and the p15 endodomain is two to three times longer than the C-terminal cytoplasmic tails of the other orthoreovirus FAST proteins (Figure 4D). The absence of significant sequence conservation and dramatically different structural features of p15 relative to p13/p14, coupled with the more distant evolutionary relationships between the parental viruses encoding these proteins, suggests p15 may have arisen by an independent gain-of-fusion event (Figure 3A). Additional sequenced isolates of these interrelated orthoreoviruses are needed in order to more robustly test this hypothesis.

The concept of independent gain-of fusion regarding p13/p14/p15 is not intended to exclude the possibility of recombination leading to acquisition of a shared non-fusogenic ancestor followed by an independent, post-speciation gain-of-fusion capability. For example, we previously speculated that FAST proteins may have evolved from viroporins or other small, non-structural, non-fusogenic viral proteins that contain diverse membrane remodeling motifs (e.g., transmembrane domains, polybasic motifs, amphipathic helices) [2,73,74]. In the above discussion, a single recombination event could have added an ORF encoding a non-fusogenic, viroporin-like FAST protein precursor to the ancestral fiber-encoding genome segment and this precursor independently acquired fusion capability and evolved to become the p13/p14 and p15 FAST proteins.

### 3.5. Novel Muscovy Duck Reoviruses Lost Fusion Capability Due to Post-Speciation Substitution Events in an Essential Multimerization Motif

While there have been reports of fusogenic MdRV isolates [47], there are no recent reports of sequenced isolates that induce syncytium formation. As noted previously [75,76], the MdRVc p11 protein is not a FAST protein homologue, implying that the loss of MdRVc fusion capacity reflects a post-speciation recombination event that replaced the ARV p10 FAST protein ORF with a heterologous ORF (Figure 3A). This is not the case for MdRVn, which clearly encodes a p10 FAST protein homologue, sharing the primary sequence conservation and almost all the hallmark features of a p10 FAST protein. The exception is a nine-residue ectodomain conserved motif (CM; AGGDLQATS) present in all other ARV and NBV p10 sequences (Figure 5A). The MdRVn isolates have two sequence variants of the CM with two to three of the first four residues matching the consensus sequence (AhGnIDSYT and AsGdIDSYT: boldface indicates sequence conservation with consensus motif; lowercase indicates MdRV sequence variants). The residues flanking the CM are also conserved between ARV and MdRV p10 proteins (Figure 5A). It seems most likely that a point substitution occurred in the CM after the classical and novel MdRVs diverged, destroying p10 function and allowing the remaining sequence to rapidly devolve from the consensus. Whatever selective pressure there might be to acquire and maintain a FAST protein, it did not prevent two distinct genetic events that resulted in the loss of fusion capability in MdRVs. The fact that MdRVn has retained almost all the hallmark FAST protein features but is non-functional for syncytium formation suggests the loss of fusion capacity may have been a recent genetic event, consistent with earlier reports of fusogenic MdRVs [47].

We further explored the hypothesis that the loss of the CM is solely responsible for the absence of MdRVn’s syncytiogenesis capacity. To generate syncytia, p10 proteins need to traffic to the plasma membrane in the correct N_out_/C_in_ membrane topology. Transfected cells expressing N-terminal FLAG-tagged versions of ARV and MdRVn p10 were treated with membrane impermeable Sulfo-NHS-LC-Biotin to label the externally exposed lysine sidechains of plasma membrane proteins, and the Western blots of the cell lysates were probed with anti-FLAG or neutravidin. As shown (Figure 5C), both proteins were expressed and detected on the cell surface at similar levels. Functional p10 proteins also need to form an intramolecular disulfide bond between the two conserved ectodomain cysteine residues in order to generate an 11-residue cystine noose fusion peptide (Figure 5A), and the formation of this structure is highly sensitive to substitutions within and adjacent to the noose [37,51,67]. A previously described assay applying a membrane-impermeable biotinylation reagent (maliemide-PEG2-biotin) to biotinylate free thiol groups on the cell surface [51,52] determined that MdRV p10 forms a cystine noose; plasma-membrane localized MdRV p10 was only labeled if cells were treated with DTT prior to the thiol-specific biotinylation reagent (Figure 5D). These results confirm that the formation of the p10 intramolecular disulfide bond occurs independent of the CM [52] and that the null syncytiogenic phenotype of novel MdRVs is not due to aberrant protein expression, trafficking, or formation of the essential cystine noose fusion peptide structure.

In ARV and NBV p10 proteins, the CM governs the cholesterol-dependent, reversible assembly of p10 multimeric fusion platforms in the plasma membrane [52]. Single, conservative point substitutions in this motif, which presumably provides a p10 multimerization binding interface, ablated multimer formation. To determine whether this motif serves the same function in MdRV p10, and whether the absence of this motif is solely responsible for the cell–cell fusion defect in novel MdRVs, we constructed a recombinant MdRV p10 (Md-CM) containing the nine-residue conserved motif (i.e., AHGNIDSYT was converted to AGGDLQATS) (Figure 5A). A time-course analysis of the syncytium formation revealed that the replacement of the CM restored MdRV syncytiogenesis to robust levels with only slightly reduced kinetics relative to ARV p10 (Figure 6A,B). Furthermore, cell-surface immunofluorescence microscopy, using cells co-transfected with N-terminally FLAG-tagged and N-terminally Myc-tagged p10 constructs, indicated that MdRV p10 failed to form the dual-color cell surface puncta that are typical of ARV p10 fusion platforms, but replacement of the CM in Md-CM p10 resulted in the extensive punctate colocalization of recombinant p10 proteins (Figure 6C).

To confirm the punctate staining pattern of the Md-CM p10 reflected assembly of multimeric fusion platforms, we employed *in cellula* FRET, as previously described [52,77]. Briefly, ARV, MdRV, and Md-CM p10 proteins were C-terminally tagged with either EGFP or mCherry. Donor and acceptor spectral bleed-through (SBT) values and normalized FRET (NFRET) intensities were calculated using the PixFRET Image-J plug-in and pixel-by-pixel analysis of sensitized emission FRET [53], and mean NFRET (mNFRET) values were determined for 10 cells in each of the two separate experiments. As shown in the FRET images (Figure 7A) and by mNFRET quantification (Figure 7B), ARVp10 formed the expected homomultimers in cells but MdRVp10 failed to multimerize. Replacement of the CM in MdRV p10 generated positive FRET signals equivalent to those observed with ARV p10, indicating that point substitutions leading to the loss of a multimerization motif required for the fusion platform assembly destroyed the ability of novel MdRVs to induce syncytium formation.

### 3.6. Final Conclusions

Current analyses support the concept that multiple recombination events were primary evolutionary drivers of the constellation of polycistronic orthoreovirus genome segments and directly contributed to the diversity of their encoded fiber and FAST proteins. The four “classes” of extant orthoreovirus fiber proteins most likely arose via three polymorphic internal recombination events in a genetically tractable region of the fiber ORF that encodes a structurally malleable repeat motif (Figure 2B). Genome segment reassortment may have influenced early inter-species evolution and the precise order of events, but nonhomologous copy choice recombination events are the most plausible explanation for the addition of 5’-extensions that generated FAST protein-encoding polycistronic genome segments. One such event led to the tricistronic, p10-encoding ARV/ARVN/NBV clade, with a subsequent post-speciation recombination event leading to the heterologous replacement of the p10 FAST protein ORF and the loss of fusion capacity by MdRVc (Figure 3A). A second independent genetic exchange led to the p14-encoding RRV/RRVT/MaRV clade and likely the related p13-encoding BrRV. There is some evidence to suggest that BRV may have acquired the p15 precursor from a third independent recombination event. At some point, a 3’-proximal genetic exchange generated the BrRV/MaRV/BRV bicistronic S4 genome segments that have the entire fiber ORF replaced with a heterologous p16 ORF of no known function. Lastly, we show that the extant MdRVs lost fusion capacity via two independent post-speciation genetic events: the recombinant loss of the entire p10 FAST protein ORF by MdRVc and point substitutions in the MdRVn CM that ablated multimeric fusion platform assembly. Together, these results highlight the genetic tractability of the orthoreovirus polycistronic genome segments and the underappreciated role of recombination in dsRNA virus evolution and the evolution of the FAST protein family.

## Figures and Tables

**Figure 1 viruses-12-00702-f001:**
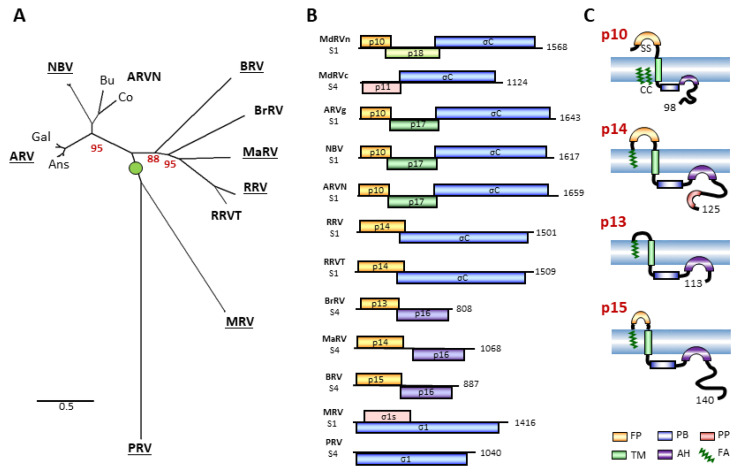
The diversity of orthoreovirus species, their polycistronic genome segments, and their fusion-associated small transmembrane (FAST) proteins. (**A**) Maximum likelihood unrooted phylogram based on 23 concatenated amino acid sequences of the seven consistently homologous structural proteins (core shell, core RdRp, core turret, core NTPase, core clamp, outer-shell, outer-clamp) encoded by the indicated 10 species or proposed species of orthoreoviruses (in boldface). See Appendix A for a list of viruses, proteins, and accession numbers. Lowercase labels denote isolates within a species: Gal and Ans are landfowl and waterfowl isolates of avian orthoreovirus (ARV); Bu and Co are bulbul and corvid isolates of neoavian orthoreovirus (ARVN). Bootstrap values (500 replicates) are shown for branch points where the associated taxa clustered together in less than 100% of the replicate trees. The tree with the highest log likelihood is shown. Branch lengths are to scale and measured by the number of substitutions per site. (**B**) Diagrams of the polycistronic S1 and S4 genome segments from the indicated orthoreovirus species, drawn to approximate scale. Open reading frames are depicted by rectangles, labeled to indicate the protein product and color-coded to depict functional or evolutionary protein relationships: yellow—FAST protein; blue—fiber protein; green—potential cell cycle inhibitory protein; purple—homologous proteins with no defined function; pink—nucleocytoplasmic shuttling protein. Numbers refer to mRNA length. (**C**) Diagrams of the orthoreovirus FAST proteins, depicting their topology in the plasma membrane. Structural motifs contained within their N-terminal ectodomains and C-terminal cytoplasmic endodomains are color-coded as described in the legend at the bottom. Numbers indicate the number of residues in each protein.

**Figure 2 viruses-12-00702-f002:**
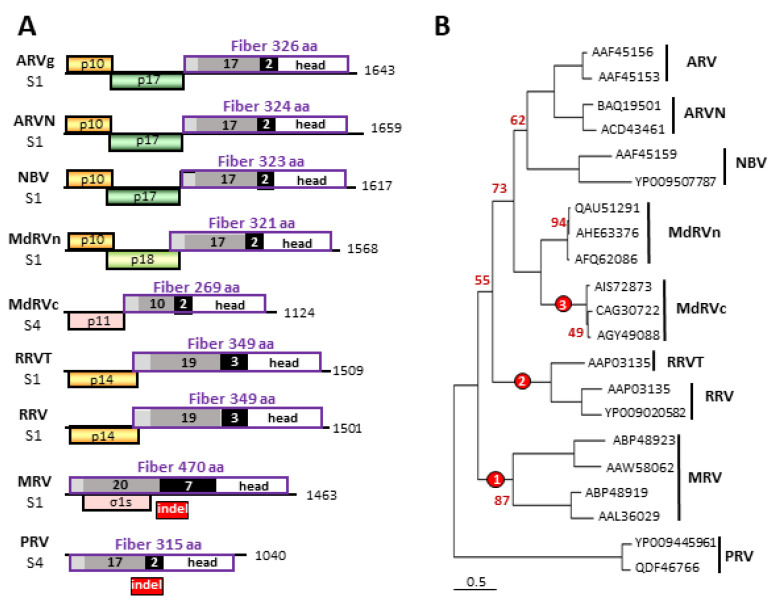
Phylogenetic analysis of orthoreovirus fiber proteins and comparisons of their polycistronic genome segments. (**A**) Diagrams of the fiber encoding S1 and S4 genome segments from the indicated orthoreovirus species, drawn to approximate scale. Numbers refer to mRNA length. Open reading frames are depicted by rectangles, labeled to indicate the protein product and color-coded to depict functional or evolutionary protein relationships: yellow—FAST protein; green—potential cell cycle inhibitory protein; pink—heterologous non-fusogenic protein of no defined function; purple outline—fiber protein. Shading within the fiber ORFs depicts the four main structural features of the fiber protein: N-terminal anchor region (light grey), heptad repeat region (dark grey), β-spiral repeats (black), and globular head (white). Numbers embedded in the shaded areas indicate the number of predicted or demonstrated heptad or β-spiral repeats. The location of the polymorphic indel region is indicated below the *Mammalian orthoreovirus* (MRV) and *Piscine orthoreovirus* (PRV) fiber protein ORFs. (**B**) Maximum likelihood unrooted phylogram of the fiber proteins of the seven indicated orthoreovirus species or proposed species and the MdRVn and classical Muscovy duck reovirus (MdRVc) subgroups of waterfowl ARV isolates. The tree with the highest log likelihood is shown and included 21 amino acid sequences. The virus isolates used for the analysis are indicated by their accession numbers. Bootstrap values (500 replicates) indicating the percentage of trees in which the associated taxa clustered together in <95% of the replicate trees are shown next to the branches. Branch lengths are to scale and measured in the number of substitutions per site. The locations of three indel events that generated the extant fiber proteins are indicated (red circles).

**Figure 3 viruses-12-00702-f003:**
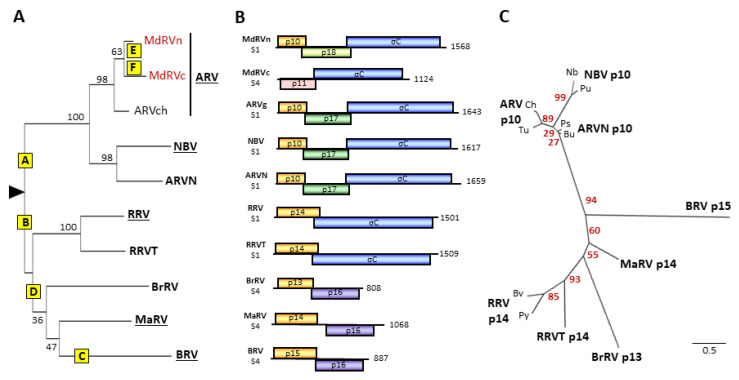
Phylogenetic analysis of fusogenic orthoreoviruses and FAST proteins and comparisons of their polycistronic genome segments. (**A**) Maximum likelihood unrooted phylogram of the sigma-class outer capsid clamp proteins of the eight fusogenic orthoreovirus species. The tree with the highest log likelihood is shown and included 10 amino acid sequences. See Appendix A for a list of viruses, host species, and accession numbers The ARV species included single isolates from three subgroups, chickens (ARVch) and classical (MdRVc) and novel (MdRVn) MdRVs. Bootstrap values (500 replicates) indicating the percentage of trees in which the associated taxa clustered together are shown next to the branches. Branch lengths are to scale and measured by the number of substitutions per site. Yellow boxes labeled A–F indicate predicted genetic events leading to the extant species, as described in the text: A—recombinant gain of p10 FAST protein precursor; B—recombinant gain of p13/p14 precursor; C—potential recombinant gain of p15 precursor; D—recombinant gain of p16 and loss of fiber proteins; E—point mutations ablate p10 function; F—recombinant loss of p10 FAST protein. (**B**) Diagrams of the polycistronic S1 and S4 genome segments, drawn to approximate scale, of the indicated eight species or proposed species of fusogenic orthoreoviruses, plus the MdRVc (MdC) and MdRVn (MdN) subgroups of non-fusogenic ARVs from waterfowl. Open reading frames are depicted by rectangles, labeled to indicate the protein product and color-coded to depict functional or evolutionary protein relationships: yellow—FAST protein; blue—fiber protein; green—potential cell cycle inhibitory protein; purple—homologous proteins with no defined function; pink—heterologous nonfusogenic protein of no defined function. Numbers refer to mRNA length. (**C**) Maximum likelihood unrooted phylogram of the FAST proteins encoded by the indicated eight species or proposed species (boldface) of orthoreoviruses. The tree with the highest log likelihood is shown and included 14 amino acid sequences. See Appendix A for a list of viruses, host species, and accession numbers. Lowercase labels denote isolates within a species: Ch and Tu are chicken and turkey isolates of ARV; Bu and Ps are bulbul and psittacine isolates of ARVN; Nb and Pu are the prototype NBV and Pulau isolates of NBV; Bv and Py are the bush viper and python isolates of RRV. Bootstrap values (500 replicates) indicating the percentage of trees in which the associated taxa clustered together are shown next to the branches. Branch lengths are to scale and measured by the number of substitutions per site.

**Figure 4 viruses-12-00702-f004:**
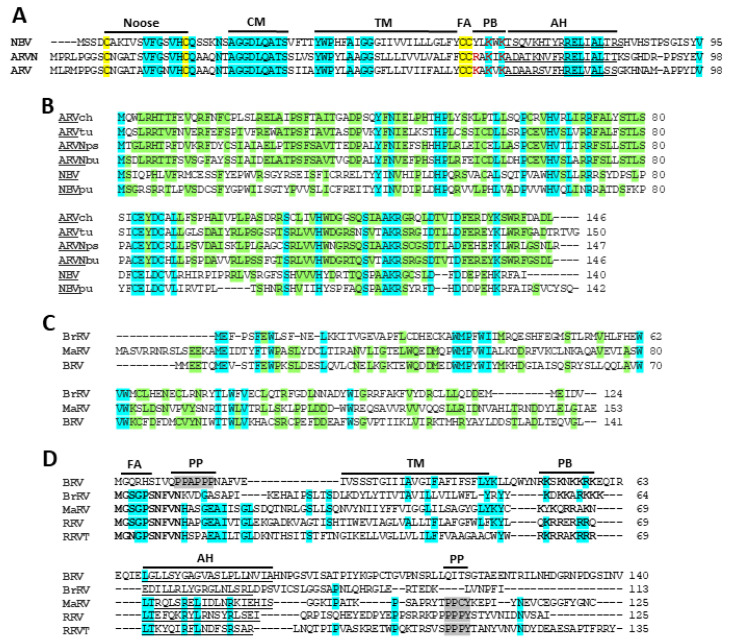
Sequence alignments and hallmark features of orthoreovirus FAST proteins and secondary polycistronic genome segment ORFs. (**A**) Sequence alignments of the p10 FAST proteins encoded by the indicated three orthoreovirus species or proposed species. Residues conserved in all three species are highlighted in cyan while the four conserved cysteines are highlighted in yellow. Hallmark p10 motifs are indicated: Noose, cystine noose formed by Cys9-Cys21; CM, conserved multimerization motif; TM, transmembrane domain; FA, palymitoylated di-cysteine motif; PB, polybasic motif with basic residues in red boldface; AH, predicted amphipathic helix underlined. (**B**) Sequence alignments of the p17 proteins encoded by the indicated fusogenic orthoreovirus species or proposed species. Lowercase labels denote isolates within a species: ch and tu are chicken and turkey isolates of ARV; bu and ps are bulbul and psittacine isolates of ARVN; Pu is the Pulau isolate of NBV. Residues conserved in all alignments are highlighted in cyan. Residues in p17 that are conserved in at least one isolate from two of the three virus species are shaded green. (**C**) Sequence alignments of the p16 proteins encoded by the indicated fusogenic orthoreovirus species or proposed species. Residues conserved in all alignments are highlighted in cyan. Residues shaded green are conserved in two of the three virus species (**D**) Sequence alignments of the p13, p14, and p15 FAST proteins encoded by the indicated five orthoreovirus species or proposed species. Alignments were generated using Clustal Muscle and manually adjusted to align the indicated hallmark structural and functional motifs of FAST proteins: FA, fatty acid myristoylation motif (GXXXS); TM, transmembrane domain; PB, polybasic motif, basic residues in boldface; AH, predicted amphipathic helix (underlined); PP, polyproline motifs include a PPXY motif in p13/p14 and a polyproline helix (PPAPPP) in p15, shaded grey. The N-terminal decapeptide conserved in p13/p14 is in boldface. Residues conserved in three or more sequences are highlighted in cyan.

**Figure 5 viruses-12-00702-f005:**
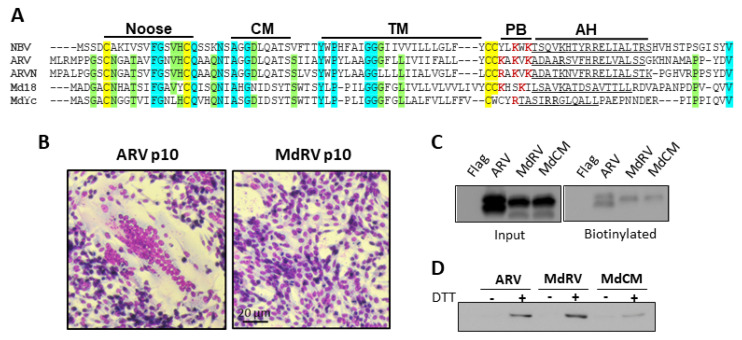
Novel MdRVp10 forms a cystine nose fusion peptide and traffics to the plasma membrane but is non-fusogenic. (**A**) Sequence alignments of the p10 FAST proteins encoded by the indicated fusogenic orthoreovirus species (Nelson Bay orthoreovirus (NBV), AF218360; ARV, AF218358; ARVN, AB914766) and two MdRVn isolates of ARV MdRVn-J18 (Md18, AFV52275) and MdRVn-Ych (MdYc, QEQ13297). The indicated hallmark p10 features are labeled as in Figure 3. The conserved cysteine residues are highlighted in yellow, completely conserved amino acids are highlighted in cyan, and residues conserved in four of five alignments are shaded green. Basic amino acids within the PB motif are in red boldface. (**B**) Transfected QM5 cells expressing ARV or MdRV p10 were fixed and Giemsa stained at 48h post-transfection to detect syncytium formation by bright field microscopy at 100x magnification. (**C**) QM5 cells expressing N-terminally FLAG-tagged versions of the indicated p10 constructs were labeled at 24 h post-transfection using membrane-impermeable Sulfo-NHS-LC-Biotin and biotinylated surface proteins were isolated using neutravidin beads and detected by Western blotting using an anti-FLAG antibody. (**D**) QM5 cells expressing N-terminally Myc-tagged versions of the indicated p10 constructs were labeled at 24 h post-transfection with membrane-impermeable maleimide-PEG2-biotin to detect free thiol groups on the cell surface, with or without prior treatment with 0.1 mM dithiothreitol (DTT) to reduce disulfide bonds. Biotinylated proteins were isolated using neutravidin beads and detected by Western blotting using an anti-Myc antibody.

**Figure 6 viruses-12-00702-f006:**
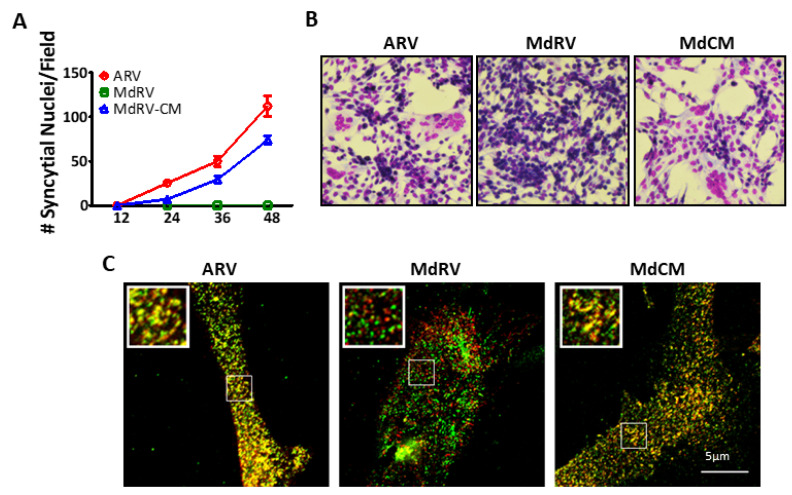
The ARV p10 CM restores fusion activity and clustering of MdRV p10 in plasma membrane microdomains. (**A**) Transfected QM5 cells with authentic ARV or MdRV p10 or expressing the recombinant MdRV/ARV p10 CM construct (MdCM) were fixed and Giemsa stained at the indicated times post-transfection, and syncytial nuclei present in five random fields from triplicate wells were quantified by bright field microscopy at 100× magnification. Results are mean ± SEM (n = 3 experiments). (**B**) Transfected QM5 cells expressing the p10 constructs described in panel (a) were fixed and Giemsa stained at 36 h post-transfection to detect syncytium formation by bright field microscopy at 100× magnification. (**C**) Transfected QM5 cells co-expressing N-terminally FLAG-tagged and N-terminally Myc-tagged versions of the indicated p10 constructs were fixed without permeabilization at 24 h post-transfection, and surface-localized p10 proteins were detected using mouse-α-FLAG and rabbit-α-Myc antisera with Alexa Fluor 488 goat anti-mouse (green) and Alexa Fluor 555 goat anti-rabbit (red) secondary antibodies. Images of the two fluorescence channels are superposed in the merged image. Boxed regions are magnified in the insets.

**Figure 7 viruses-12-00702-f007:**
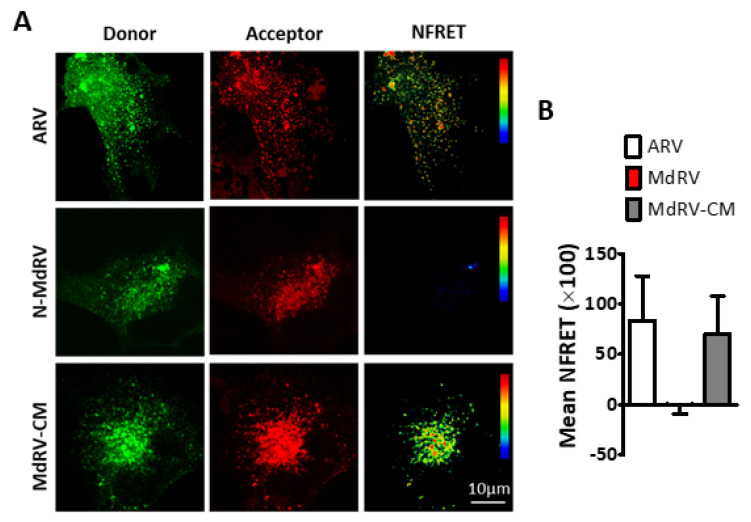
The ARV p10 CM restores MdRV p10 multimerization. (**A**) QM5 cells were co-transfected with C-terminally EGFP- and mCherry tagged versions of the indicated p10 constructs and analyzed for sensitized emission FRET. Images from the donor and acceptor channels and the calculated normalized FRET (NFRET) image are shown. The NFRET range is denoted by color gradations. (**B**) Best-fit Gaussian distributions of calculated NFRET images from 10 cells in each of the two separate experiments were used to calculate mNFRET values, presented as mean ± SEM, from cells co-transfected with fluorescently tagged versions of the indicated p10 constructs.

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
