# Peer review of "Polycistronic Genome Segment Evolution and Gain and Loss of FAST Protein Function during Fusogenic Orthoreovirus Speciation"

_viruses, 2020, doi:10.3390/v12070702_

Round 1
Reviewer 1 Report
This paper presents a sequence comparison of the polycistronic reovirus gene segments containing the fiber protein and small non-structural proteins involved in syncytiogenesis/cell-cell fusion (FAST protein) or related membrane-associated functions (e.g viroporins), indicating a mechanism of evolution through several recombination events.A specific comparison was made between two different Muscowy Duck reovirus isolate p10 sequences, assumed to have fusogenic and non fusogenic functions respectively. A recombinant expression of an MdRV p10 sequence, an ARV p10 sequence, and a motif-converted MdRV p10 was performed to demonstrate their functional differences.
The topic of reovirus evolution and the functional concequences is a very interesting one. However, although this manuscript is well written with nicely presented figures there are some major concerns
Although the manuscript appear as a thorough overview and comparison of reovirus FAST protein sequences and structural variants in a range of species it has several drawbacks regarding the link between the data and conclusions drawn. In particular, essential information is missing or confusingly presented, making it hard to evaluate some of the data.
The introduction lack some relevant background for the results. In particular, it does not refer too, or even mention, the background literature on MdRV or the biological/pathological data related to ARV, even if it is important for the study of functional differences. More information on reovirus replication cycle, structure and total genome would have lifted the introduction.
The article claim to give evidence of several recombination events leading to gain-of-fusion and loss-of fusion events. For this, the data are not sufficiently convincing. The concept of recombination in an RNA virus replicating exclusively cytoplasm is different. In addition, the concept of segment reassortment as an evolutionary concept in such a segmented RNA virus should also be discussed. The complete virus genomes (additional segments) should be pulled into the discussion as well.
The fusion proteins are aligned and compared and thoroughly discussed, but the specific sites of the recombination events are not clarified or shown, and although the data are referred to as "evidence" (ln 117, ln307) they appear as hypotheses and discussion in the text, referred to as "likely", "simplest explanation for", "presumably". The hypothesis is ok, but it is hard to locate evidence in the way sequence data are presented here. At least this must be more clearly illustrated, not only discussed in the text. Now, this is not clear, and the novelty in the evolutionary concept not convincing.
The MdRV and ARV sequences of particular importance here lack some basic information or appropriate References. Accession numbers of the cloned MdRV/ARV sequences should be given in the main manuscript (not only in the supplementary file), and the naming of the isolates should be coherent between the manuscript and supplementary table (MdC and MdN are not coherent with supplementary table names), and between the main text and figures. As an example, MdRV are just included as "waterfoul isolates of ARV" in the introduction and occurs first in the method section as the abbreviation without further explanation or specification of sequence/isolate cloned. This is not specified in the result section either. In figure 5, isolates Md03 and MdYc are referred to, but these names just appear here. It is hard to interpret if the classical or novel MdRV p10 is studied in the functional experiments.
Figure 6 and 7 provide exciting mechanistic data on sequences important for fusogenic function, but should have provided more detail on the actual clones (full aa sequence as supplementary file), figure 6A should have been supplemented with fusion images (Nuclear staining) showing changes in fusogenic Activity(like in 5B). The choice of quail cells for the experimental study should also be addressed. How could that impact results when these proteins clearly interact with proteins of their natural hosts.
The manuscript would also benefit from a better separation between introduction/background, results and discussion. Both introductory information and discussion is mixed into the results section of the manuscript, and the information under the introduction heading does not completely cover the background.
Author Response
We thank the reviewer for their comments, most of which relate to deficiencies in how we presented and discussed our analyses and linked this information to our conclusions. As requested, we have added extensive new discussion of the background literature relative to our results, which makes the manuscript more readable and strengthens support for our conclusions and inferences based on our analyses. These additions are described below in response to specific comments and suggestions.
- Several comments were in regard to the lack of suitable background information in the introduction (“background literature on MdRV or the biological/pathological data related to ARV”; “ More information on reovirus replication cycle, structure and total genome would have lifted the introduction").
> We agree this background information would be useful to readers and could better introduce the underexamined concept of recombination, not just genome segment reassortment, as a driver of reovirus evolution.
> We have made extensive additions to the Introduction to include discussion of the reovirus replication cycle, genome organization, virion structure, RNA transcription and replication, and how this relates to genome segment reassortment (ln 73-92).
> We included more detailed discussion of the structure of the reovirus fiber protein and how it varies between virus species due to internal deletions in the fiber domain of the fusogenic reoviruses. This background information is linked to our analysis showing that evolution of the fiber-encoding genome segment also involved a 5’-teminal extension of the fiber-encoding genome segment (ln 93-105).
> We have also added a paragraph to introduce the classical and novel Muscovy duck reovirus (MdRV) subgroups and the limited discussion in the literature on their polycistronic genome segments (ln 151-164). We agree that providing this information earlier improves the readability of the manuscript and better aligns with the flow of the figures (a suggestion for improvement from Reviewer 2).
- “The concept of recombination in an RNA virus replicating exclusively cytoplasm is different (and) the concept of segment reassortment as an evolutionary concept in such a segmented RNA virus should also be discussed”
> As noted, the concept of recombination as a driver of reovirus evolution has not been widely considered or examined, with point mutations and genome segment reassortment being the most extensively characterized genetic events. We agree with the novelty of the concept regarding reovirus evolution and have added considerably more reference to the relevant literature on homologous and nonhomologous RNA virus recombination, particularly the extensive literature on recombination with plus-strand RNA viruses (e.g., ln 333-339, ln 349-359) and a recent report on numerous predicted recombination events between co-circulating strains of avian reovirus (ln 354-357, 449-451). We cite this literature in the context of our results to support our inferences and conclusions. We also discuss predicted recombination events with reoviruses and how a copy choice recombination mechanism can easily explain acquisition of a 5’-extension that provided a “pre-fusion” genetic module to the fusogenic reoviruses.
> We have added additional text to indicate the prominent role of genome segment reassortment in reovirus evolution (ln 87-92).
> We have condensed and modified previous section 3.5 (now a short paragraph to finish section 3.4, ln 509-517) to refocus this discussion on clarifying for readers that the concept of independent gain-of fusion regarding is not intended to exclude the possibility of recombination leading to acquisition of a shared nonfusogenic ancestor followed by independent, post-speciation gain-of-fusion capability. This change was also requested by Reviewer 2.
- “The complete virus genomes (additional segments) should be pulled into the discussion as well”
> We are not sure what is requested. We included a phylogram of the concatenated seven homologous structural proteins from 23 virus isolates covering all 10 species or proposed species (Figure 1A), and also phylograms of the outer clamp proteins and FAST proteins of the fusogenic species (Figures 3A and C). We now include phylograms of the fiber protein (new Figure 2B) to expand our analysis of the evolution of the polycistronic genome segments.
- “The specific sites of the recombination events are not clarified or shown”
> As mentioned in the text, the limited number of sequences available for several of the viruses restricts our ability to define specific recombination sites. As we now discuss in more detail (ln 357-362), the inferred recombination events that extended the 5’-end of the S1 genome segments were most likely nonhomologous, meaning the RNA was acquired from another genome segment or another virus or cellular RNA, and these events were likely ancient occurring as fish, reptile and avian reoviruses diverged with their hosts. Combined with the very high degree of sequence divergence of all the polycistronic genome segment ORFs, defining specific recombination sites, even with a much more comprehensive sequence analysis, may not be achievable. We agree the recombination events we describe are inferred, but strongly supported, by our analysis.
- “Accession numbers of the cloned MdRV/ARV sequences should be given in the main manuscript; naming of the isolates should be coherent between the manuscript and supplementary table (MdC and MdN are not coherent with supplementary table names; Fig 5 Md03 and MdYc are referred to)”
> We thank the reviewer for noting these oversights and discrepancies. We have checked the manuscript for virus nomenclature consistency. We also provided a more complete list of accession numbers for each of the relevant figures but feel this is best provided supplementary tables, not in the figure legends that would become overly lengthy (e.g., 7 proteins from 23 viruses were used for Figure 1A; 21 viruses were used for Figure 2B).
- “Provide more detail on the actual clones (full aa sequence as supplementary file), figure 6A should have been supplemented with fusion images (Nuclear staining) showing changes in fusogenic activity (like in 5B)”
> We have added syncytia images to Figure 6 as requested. We are unsure what information is requested and for what clones. We have provided the full amino acid sequence of the parental p10 FAST proteins from ARV and MdRVn (Figure 5A) and stated the 9-residue sequence of the ARV CM that was used to replace the corresponding sequence in MdRV.
- The choice of quail cells for the experimental study should also be addressed. How could that impact results when these proteins clearly interact with proteins of their natural hosts.
>We have not tested the constructs in duck cells or chicken cells but all of the FAST proteins readily fuse QM5 quail cells. As we show here, that is not the case for MdRV unless we replace the missing multimerization motif. In this regard, the QM5 cells provide a suitable readout for fusion competency.
- Better separation between introduction/background, results and discussion
> We have reorganized the combined Results and Discussion section and some of the figures and figure panels to improve the flow of the data and text. We also added three paragraphs to the Introduction to provide more relevant background on the reovirus replication cycle and genome organization and reassortment, and we moved the Muscovy duck reovirus (MdRV) background from the Results and Discussion to the Introduction.
Reviewer 2 Report
FAST proteins are small nonstructural proteins encoded by members of the orthoreovirus and aquareovirus genera that mediate cell-cell fusion and syncytium formation. When present in the viral genome, FAST proteins are encoded by ORFs embedded in bi- or tri-cistronic mRNAs. Previous studies analyzing six reovirus species suggested extant orthoreovirus and aquareovirus FAST ORFs arose through two separate gain-of-fusion events by a common ancestral, nonfusogenic reovirus. In the current manuscript, Yang et al., phylogenetically analyzed sequences encoding FAST proteins and other accessory proteins from at least ten identified or predicted reovirus species to provide evidence for at least two independent gain-of-fusion events and a post-speciation loss-of-fusion event by extant fusogenic orthoreoviruses. The authors expressed recombinant versions of FAST proteins in avian cells and used techniques, including FRET, to demonstrate that a nine-residue multimerization motif mediates fusion capacity in one of two subgroups of waterfowl avian reovirus isolates. Together, these results suggest a complex evolutionary history from a common non-fusogenic ancestor to generate the diversity of FAST proteins detected today in orthoreoviruses and aquareoviruses.
Broad comments:
FAST proteins are fascinating molecules that influence the virulence of the Reoviridae family viruses that encode them, and these viruses infect a broad range of hosts. Tracing the history of accessory genes, like those that encode the FAST proteins, in Reoviridae genomes may provide insight into general mechanisms of virus evolution. Thus, the current study is of broad interest for evolutionary biologists as well as reovirologists. The specifics of the FAST proteins and their putative evolutionary history are complex, but the authors explain concepts clearly enough to follow the reasoning. Since sequences of new and predicted reoviruses have been included in the phylogenetic analyses, the current study provides new information that builds on the previous study. The authors’ conclusions are generally supported by the data. However, analysis or discussion of phylogenetic relationships among the fiber proteins of viruses within and outside the ARV/NBV clade and may lend strength to some of the authors’ arguments regarding the timing of speciation and acquisition events. Similarly, although the authors mention that for some species examined only a single representative sequence is available, phylogenetic analyses would be strengthened by the inclusion of additional sequences when possible. Data from experiments conducted in cells expressing recombinant forms of FAST proteins are well rationalized and convincing. The 9-residue motif in p10 does appear to function as a multimerization platform, and that its alteration alone restores fusion activity suggests that its mutation has resulted in loss of function among its subgroup of new waterfowl avian reovirus isolates.
Specific comments:
- It is somewhat distracting that figure panels are discussed out of order in the text (e.g., 2A-B and 3B are mentioned before 2A or 3A is ever mentioned). Consider reorganizing so panels are presented in the order they are discussed.
- Section 3.5 is interesting but highly speculative without data from the current study seeming to support the speculations. Consider removing this section or further discussing how the new data shape these hypotheses.
- What is the presumptive pressure for new MdRVs to retain the nonfusogenic p10 ORF? Some discussion of this point would be interesting.
- Lines 315-317 present the only obvious discussion of how and why so many distinct putative recombination events occurred at the same location in the viral genome. Please discuss this issue further. Is this surprising or typical? Can you provide examples of other viruses for which multiple, distinct evolutionary events have been reported at a single site? Are there any other sites in Reoviridae genomes where variability/evolutionary events commonly occur?
Author Response
- Analysis or discussion of phylogenetic relationships among the fiber proteins of viruses within and outside the ARV/NBV clade and may lend strength to some of the authors’ arguments regarding the timing of speciation and acquisition events.
> We thank the reviewer for this suggestion and complied. We have included more extensive discussion of the reovirus fiber proteins and their structural features in the Introduction (ln 93-105) and tied this background information to new discussion of relationships between the different fiber proteins of the reovirus species (new subheading 3.1). New sequence and phylogenetic analyses (new Figure 2 and associated text, ln 291-341) highlights a hotspot for polymorphic internal indels in the fiber ORF. More notably, and as suggested by the reviewer, this analysis does help with the trajectory of events by suggesting 5’-terminal extensions to a piscine reovirus-like ancestral fiber gene as the most plausible explanation for evolution of the polycistronic genome segments and FAST proteins.
- Phylogenetic analyses would be strengthened by the inclusion of additional sequences when possible.
> There are only single sequences available for four of the eight reovirus species (BRV, BrRV, MaRV and RRVT) and only two for another (RRV). Multiple isolates from the remaining three species (ARV, MRV, NBV) are highly similar within a species and highly dissimilar between species. Consequently, addition of more isolates does not significantly alter tree topologies or strengthen weak bookstrap values.
- Somewhat distracting that figure panels are discussed out of order in the text.
> New introductory material requested by Reviewer 1 now brings the figure panels and their order of referral in the text into alignment. We also rearranged some figure panels and collapsed previous Figures 3 and 4 into new Figure 4 to improve the flow.
- Section 3.5 is interesting but highly speculative. Delete or better integrate with data.
> We have deleted most of section 3.5 and modified a condensed version (now a short paragraph to finish section 3.4, ln 509-517) to focus on clarifying for readers that the concept of independent gain-of fusion is not intended to exclude the possibility of recombination leading to acquisition of a shared nonfusogenic ancestor followed by independent, post-speciation gain-of-fusion capability. We feel is it important to make this point clear.
- Discuss the presumptive pressure for new MdRVs to retain the nonfusogenic p10 ORF?
> We have added sentences (ln 535-537) suggesting the loss of fusion capacity may have been a recent genetic event, consistent with reports from the 1980’s of fusogenic MdRVs, which suggests the nonfusogenic ORF may not be maintained over time.
- Why do so many distinct putative recombination events occurred at the same location in the viral genome? Is this surprising or typical? Can you provide examples of other viruses for which multiple, distinct evolutionary events have been reported at a single site? Are there any other sites in Reoviridae genomes where variability/evolutionary events commonly occur?
> We do not know why the 5’-terminal region of an ancestral monocistronic fiber ORF might have been the target for the three inferred recombination events. We have referenced a recent paper with avian reoviruses showing evidence for multiple homologous recombination events between co-circulating strains (ln 354-357, ln 449-451). We have also added discussion of recombination events in other RNA viruses and referred readers to reviews of extensive literature with plus-strand RNA viruses, including reference to recombination hotspots that are influenced by RNA secondary structures (ln 337-339). It may be that RNA secondary structures near the 5’-termini promoted these orthoreovirus recombination events.